# InsectOR—Webserver for sensitive identification of insect olfactory receptor genes from non-model genomes

**Snehal Dilip Karpe** ¤ᵃ¤ᵇ, **Vikas Tiwari, Sowdhamini Ramanathan** *

National Centre for Biological Sciences (NCBS), TIFR, Bengaluru, Karnataka, India

¤a Current address: Laboratory of Experimental Hematology, Institut Jules Bordet, Université Libre de Bruxelles, Brussels, Belgium
¤b Current address: Unit of Animal Genomics, GIGA, University of Liège, Liège, Belgium
* mini@ncbs.res.in

**Data Availability Statement:** The insectOR webserver can be accessed at "http://caps.ncbs.res.in/insectOR/". The core package for standalone use can be downloaded from "https://github.com/sdk15/insectOR.

## Abstract

Insect Olfactory Receptors (ORs) are diverse family of membrane protein receptors responsible for most of the insect olfactory perception and communication, and hence they are of utmost importance for developing repellents or pesticides. Accurate gene prediction of insect ORs from newly sequenced genomes is an important but challenging task. We have developed a dedicated webserver, 'insectOR', to predict and validate insect OR genes using multiple gene prediction algorithms, accompanied by relevant validations. It is possible to employ this server nearly automatically and perform rapid prediction of the OR gene loci from thousands of OR-protein-to-genome alignments, resolve gene boundaries for tandem OR genes and refine them further to provide more complete OR gene models. InsectOR outperformed the popular genome annotation pipelines (MAKER and NCBI eukaryotic genome annotation) in terms of overall sensitivity at base, exon and locus level, when tested on two distantly related insect genomes. It displayed more than 95% nucleotide level precision in both tests. Finally, given the same input data and parameters, InsectOR missed less than 2% gene loci, in contrast to 55% loci missed by MAKER for *Drosophila melanogaster*. The webserver is freely available on the web at http://caps.ncbs.res.in/insectOR/ and the basic package can be downloaded from https://github.com/sdk15/insectOR for local use. This tool will allow biologists to perform quick preliminary identification of insect olfactory receptor genes from newly sequenced genomes and also assist in their further detailed annotation. Its usage can also be extended to other divergent gene families.

## Introduction

Insect biology has been studied extensively over the years for human benefit–to collect honey, pollinate crops, ward off pests, etc. Recently, these diverse species are also being used as model organisms for modern experiments to understand their (and in-turn our own) biology in intricate details. Advent of Next Generation Sequencing (NGS) technologies has given us powers to study this vast diversity at genomic level [1]. Through projects like Arthropod i5k, Earth

**Funding:** This work was funded by Shyama Prasad Mukherjee Fellowship from Council of Scientific and Industrial Research (CSIR), India and later Bridging Postdoctoral Fellowship from National Centre for Biological Sciences (Tata Institute of Fundamental Research), India for SDK; and JC Bose Fellowship (JC Bose fellowship (SB/S2/JC-071/2015) from Science and Engineering Research Board, India for RS. The authors used the infrastructural facilities of National Centre for Biological Sciences (NCBS). The funders had no role in study design, data collection and analysis, decision to publish, or preparation of the manuscript.

**Competing interests:** The authors have declared that no competing interests exist.

BioGenome thousands of insect genomes and transcriptomes will be available soon and we need powerful bioinformatics tools to analyse the data [2, 3].

Efforts are underway to exploit understanding of insect olfaction to manage pests and disease vectors [4–6]. Insect olfaction is also an interesting system for study due to its commonalities and differences with the vertebrate olfactory system [7–9]. The discovery of insect olfactory receptors was itself largely dependent on the early bioinformatics analyses looking for novel protein coding regions with mammalian 'GPCR-like' properties in *Drosophila melanogaster* genome, which were further validated using antennae-specific expression [10–13]. Further OR discoveries in other genomes started to depend on their homology with the Drosophila ORs [14–16].

Later, vast differences in the average numbers and sub-families of ORs were observed across various insect orders [17, 18]. Although OR repertoires from multiple species are available today, they still remain elusive in the genome due to this diversity. Insect ORs belong to a diverse family of proteins varying across insect orders [18]. In addition, the gene models of ORs also vary from one sub-family to another e.g. various OR subfamilies within the insect order Hymenoptera uniquely possess 4 to 9 exons [19]. This leads to lack of well-curated OR queries for use within general genome annotation pipelines. These automated genome annotations usually start with de novo gene predictions followed by homology-based corroborations. Probably, as these pipelines are trained on only one or few model organism annotations before use, they fail to capture the entire OR gene repertoire in an insect genome. Our previous work has shown that only 60–70% of the total OR gene content is recovered by the general gene annotation pipelines [19, 20]. ORs are mostly selectively expressed in antennae, differ from one insect order to another and undergo rapid births and deaths as per the requirements of each species, which causes missing and miss-annotations in the de novo and homology-based gene prediction of these genes. Hence, special efforts (e.g. antennal transcriptome sequencing or extensive manual curation) are necessary to detect insect ORs with good sensitivity and precision [21].

Some of these problems could be alleviated by giving preference to homology-based gene predictions. In spite of that, we may find faulty gene predictions. ORs are usually present in tandem repeats in insect genomes and the alignments with OR protein queries may span two different gene regions and give erroneous gene predictions. This can also lead to miss-annotation of the gene and intron-exon boundaries. This problem could be addressed by transcriptome sequencing of the antenna, which is often costly and dependent on the availability of the antennae samples. It is also most likely to not cover the entire OR gene repertoire in cases of time-dependent/exposure-dependent expression of the OR genes [22]. Pipelines like OMIGA [23] are dedicated for insect genomes, but require transcriptome evidence to recognize OR genes. Hence most insect genome assembly and annotation projects are followed up by time-consuming, further experimental data or laborious homology dependent manual curation of ORs. To the best of our knowledge, currently there is only one recently developed, dedicated pipeline or webserver for prediction of genes from a single protein family as diverse as insect olfactory receptors, however it has been tested on the Niemann-Pick type C2 (NPC2) and insect gustatory receptor (GR) gene families and not olfactory receptors [24]. Hence a pipeline, with simplified and specific search for this OR family, without incorporating problems of general genome annotation pipelines, is of great value to the ever-growing insect genomics community.

We developed such a computational stand-alone pipeline during annotation of ORs from two solitary bees [20]. We have improved it further, added modules to assist automated refining and validation of genes and we are presenting it here in the form of a webserver, insectOR. Redundant hits are filtered, starting from alignment of multiple ORs to the genome of interest, to provide sensitive prediction of OR gene models.

## Methods

### Input parameters

Exonerate alignment file with additional Generic Feature Format (GFF) annotations [25] generated from insect genome of interest is a mandatory input. The related FASTA files of genome and OR proteins are also necessary for better refinement of the roughly predicted gene models. The choice of the best protein queries for this search is a crucial step that can be better addressed by the user with the help of directions given on the 'About' page of the webserver and hence it is currently not automated. This also reduces the resources spent on performing Exonerate on the webserver. More directions on how to run exonerate can also be found at the 'About' page of insectOR.

Users can also choose to provide genome annotation from any other source (GFF format) for additional comparisons with insectOR predictions. One can additionally choose to perform validation of the predicted proteins using HMMSEARCH [26] against 7tm_6, the Pfam [27, 28] protein family domain which is characteristic of insect ORs. The presence or absence of the 7tm_6 domain is recorded. Users may also choose one or more of the three trans-membrane prediction (TMH) methods–TMHMM2 [29, 30], HMMTOP2 [31, 32] and Phobius [33]. If all three methods are selected, additional Consensus TMH prediction is performed [34]. InsectOR provides an option to perform additional annotation using known motifs of the insect ORs with the help of MAST tool from the MEME motif suite [35, 36]. Users can search for default set of 10 protein motifs predicted for *A. florea* ORs [19] or they may upload their own motifs of interest.

### Output

Statistics on the total number of predicted genes/gene fragments, complete and partial genes, gene regions with and without putative start sites and pseudogenous/normal gene status are provided in the final summary of the output (Fig 1A). Additionally, details of the genes encoding proteins with 7tm_6 domains are provided. Novel OR gene regions annotated by insectOR that are absent in the user-provided gene annotations are also counted. The details of each predicted OR gene can be studied from the table available in the next tab (Fig 1B). The gene predictions are displayed in the Dalliance web-embedded genome viewer [37] (Fig 1C). In case annotations from any other source are provided they are also displayed in the genome viewer and trimmed version of GFF file overlapping with insectOR prediction is available for download. Dalliance displays results in a customizable manner for easy comparison with user-provided gene annotations. Fig 1C illustrates, user-provided genes from NCBI GFF file. Zooming in onto particular regions gives more information on the coding nucleotides and the protein sequence translated by them. For the predicted OR gene regions from insectOR, final gene structure is reported in GFF and BED12+1 format and the putative CDS/transcript and protein sequence are also provided, all of which are available for download. One may use the GFF/BED12+1 formatted output/s on one of the various genome annotation editing tools (like Artemis [38], Ugene [39], Web Apollo [40] etc.) for further manual curation and editing of these genes. The gene regions with the status of 'partial' or 'pseudogenous' or 'without start codon' can be particularly targeted for curation. If user chooses to perform TMH validation by any of the three third-party methods mentioned before, a bar-plot representing the distribution of number of helices predicted by each selected TMH prediction method is plotted (Fig 1D). If all the three are selected, consensus TMH [34] is predicted and insectOR provides details of the four TMH predictions in a new result tab (Fig 1E). In case motif search is selected, the results are available at the last tab (Fig 1F).

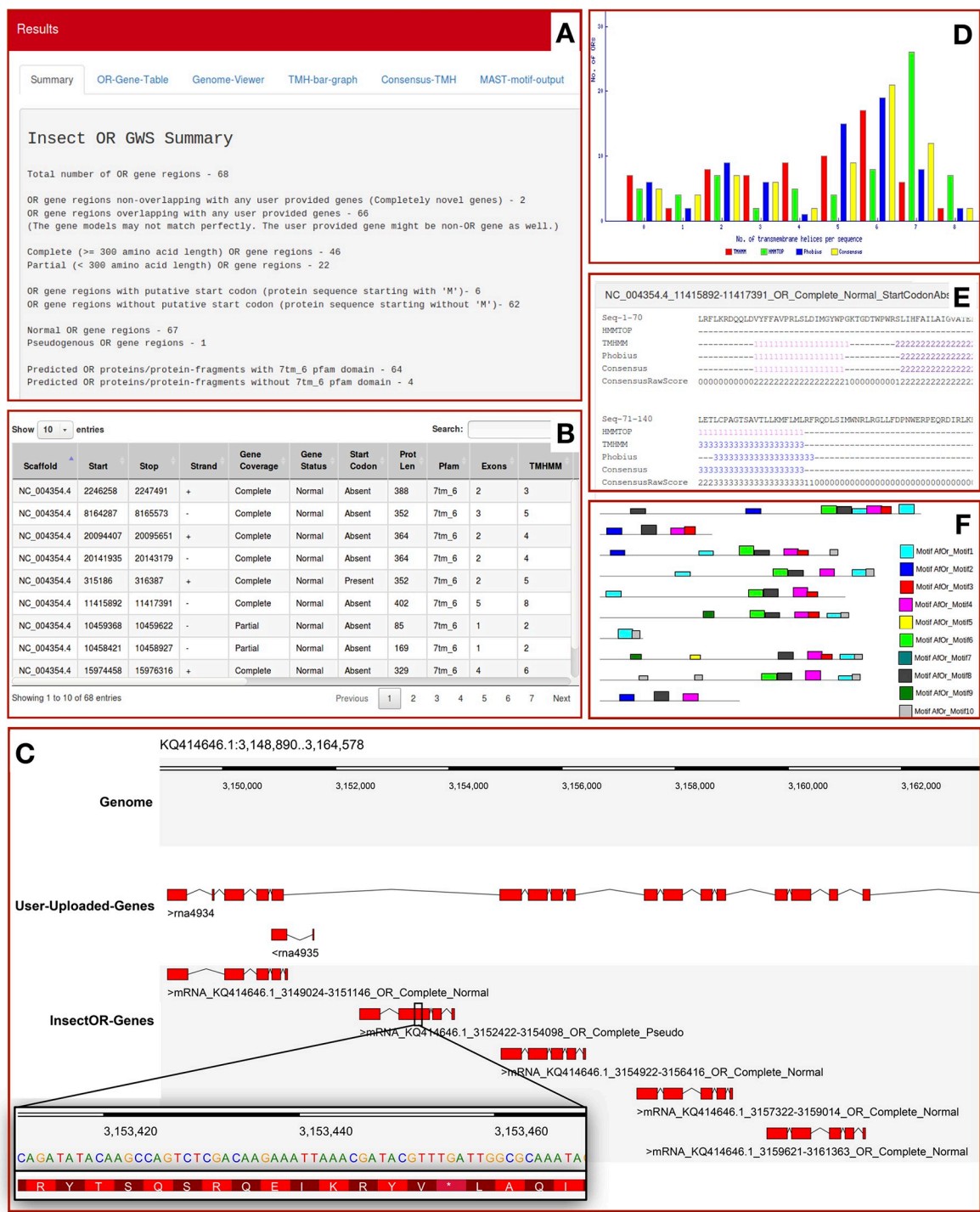

**Fig 1. A sample output from insectOR.** Section A-B and D-F are outputs derived for input Exonerate alignments for *Drosophila melanogaster* whereas section C displays information derived from *Habropoda laboriosa* alignments. Two or more of these sections are available in the output depending on the analysis chosen by the user.

## Annotation algorithm

Core annotation algorithm is written natively in Perl. It also invokes several other tools as mentioned in the 'Input' section. This algorithm processes the Exonerate alignment data to sensitively predict the OR coding gene regions and also performs validations as discussed next (Fig 2). The problem of missing and mis-annotation of tandemly repeated OR genes is addressed using 'divide and conquer' policy as described below.

Initially, OR protein-to-genome alignments are identified on the genome as follows. The exonerate output is read for each alignment. For every new genomic scaffold (target in the alignment), a virtual scaffold with the similar length with score '0' at each nucleotide position is created. Subsequently for each alignment, the score at every corresponding nucleotide position is incremented by one. This leads to virtual subalignments of OR protein-to-genome alignments demarcated by islands of higher scores (rough OR loci) on the base string of

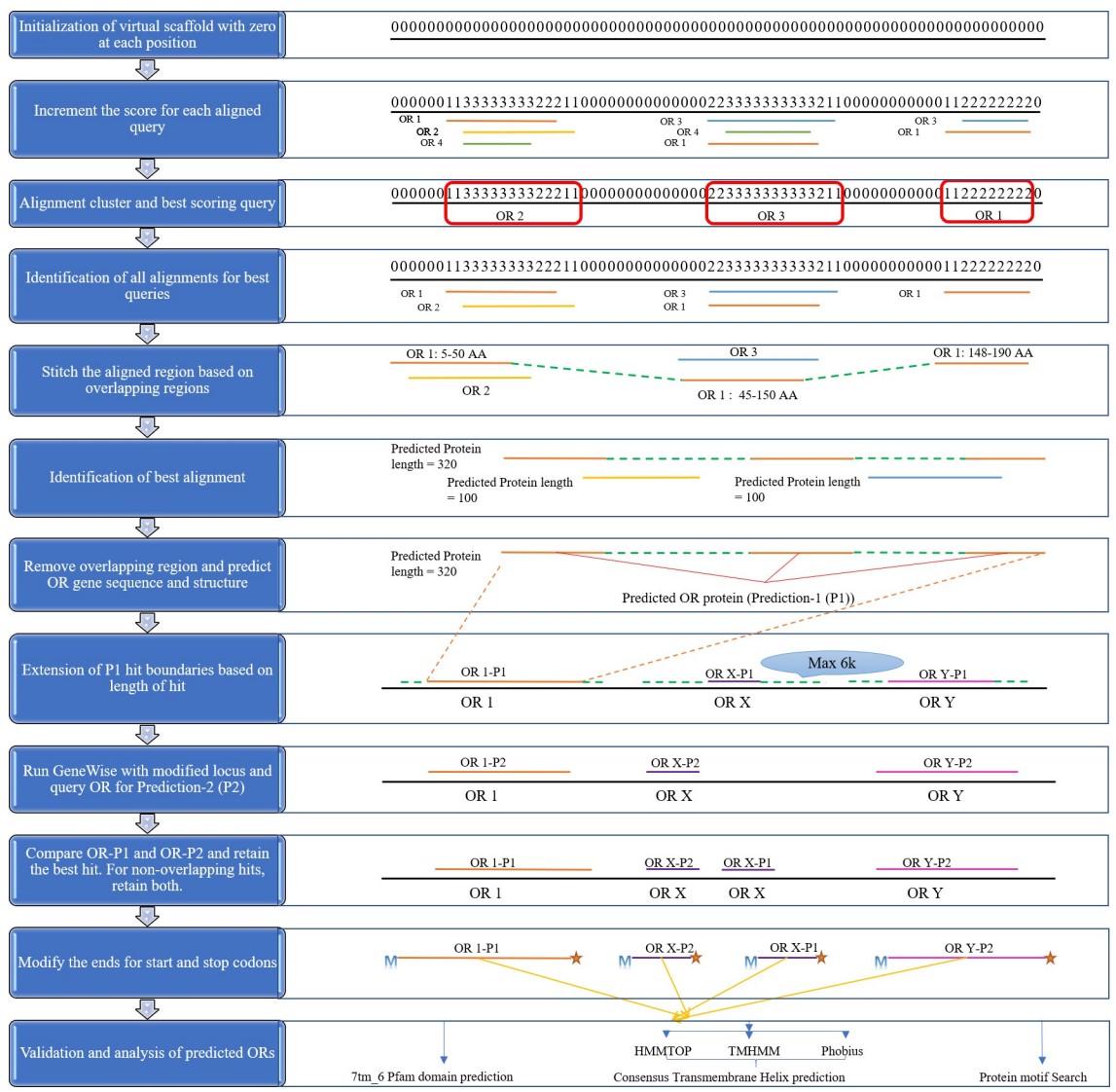

**Fig 2. Annotation algorithm part of the insectOR webserver.** Steps in the annotation algorithm are displayed here in cartoon representation.

repeated '0' scores (non-'OR' loci). As stringent cutoffs are advised for the allowed intron lengths while performing Exonerate alignment (e.g. 2000 nucleotides or less), this step helps to distinguish ('divide') between tandem OR genes in the form of closely situated but distinct alignment islands/clusters.

This is followed by the next step of selecting the best alignment/gene model for a set of sub-alignments. The sub-alignments may sometimes be too short to include full length gene alignments due to stringent intron length cutoffs. Such smaller alignment regions correspond to fragments of gene models. To resolve this, initially, the best alignment per set is selected based on the Exonerate alignment score. Corresponding query proteins for each of these best alignments in each cluster are identified as the best query proteins for the related clusters. For example, query protein OR2, OR3 and OR1 are shown as the best scoring queries in the alignment clusters 1, 2 and 3 from left to right in Fig 2. For the best queries selected per cluster, all other alignments on the same genomic scaffold are retained. In this way, from multiple redundant alignments, insectOR retains the best scoring alignment and also their neighbouring alignments from the same best scoring query.

Next, these best neighbouring alignments arising from each query are concatenated into complete protein alignments, if they are arranged congruently in the correct orientation and sequence on the genomic scaffold. In some cases, the boundary region in the alignments may be extended and the same region from the query may be aligned to the two different successive locations that need to be merged (as shown in the Fig 2 for query protein OR1; Amino acids 45 to 50 are aligned at two different locations on the scaffold whereas the flanking regions are different– 5 to 50 and 45 to 150). These are the cases of wrong extensions of the alignment fragments into introns. For such overlapping regions of the query, the possible exon-intron splicing sites are predicted based on the presence of 'gt' towards the 3' terminus of the previous exon (region where a protein fragment is aligned) and presence of 'ag' towards 5' terminus of the next exon (region where next protein fragment is aligned). The remaining regions are trimmed. All the possible combinations of such fragments are generated keeping the length of the overlapping region constant (e.g. In the above case of protein query OR1, there are 6 amino acids overlapping– 45 to 50. All combination of the concatenated nucleotide fragments giving rise to 18 nucleotide regions with flanking splice sites are considered). Next, the combination of splicing sites and their scores are compared to each other. The concatenated region providing the best similarity-based score on the Exonerate alignment is retained. In this way, insectOR finds the best possible splicing sites in cases of the fragmented alignments and stitches them to generate more complete alignments/gene models. In some cases, genes may possess more than one isoform that are formed by alternative splicing. In such cases, similar region of a query protein may be aligned at two consecutive locations (e.g. duplicated exons that are alternatively spliced to give different isoforms). If the overlap is less than 20% of any of the two query regions when aligned, the two hits are kept separate. In case of overlap, multiple parameters, such as completeness of the gene, higher protein length, non-pseudogenous nature and presence of START codon are examined (in that order).

For further refinement of gene boundaries, each gene/genic fragment (referred as prediction-1 (P1) hit is used as input for another gene structure prediction tool called "GeneWise". GeneWise is known to perform well for one-to-one protein-to-DNA alignments [41]. The genomic locus of each P1 hit is allowed to extend on either side depending upon the length of the hit and maximum boundary extension of 6000 bp. This empirical cut-off was provided based on the average intergenic region observed for multiple insect genomes. Along with the extended genomic locus sequence, the best aligned query for that region (determined earlier) are given as input to GeneWise. For each P1 hit, corresponding predicted (P2) hits are generated by running GeneWise. Further, for each locus, both P1 and P2 hits are compared. If P1

and P2 hits are overlapping, then the best of two is retained and otherwise both the hits are retained. Final hit is modified by locating the START and STOP codons (20 amino acids) upstream or downstream of the current start and end of the alignment and it is finally assigned a name according to its genomic location. Also, the presence of ATG (start codon) at the N-terminus and pseudogenizing elements (frameshifts or stop codons with respect to the query protein) are noted and included in the gene name. Based on the user-provided completion cut-off (default: 300 amino acids), a genic region is either declared as complete or partial.

In the last step of the pipeline, various validations on the predicted protein sequences are performed. Although TMH prediction programs are not very accurate (and may predict less or more than 7 helices for an insect OR), the presence of at-least few TMHs (depending on the protein fragment length) is necessary for validation. More robust validation comes from the search for '7tm_6' domain. Users may also choose to scan for protein motifs of interest in the predicted proteins. With more ongoing research on insect ORs, presence or absence of certain OR protein motifs may provide affirmation of their specific insect order origin [42] and might also provide clues regarding the kind of the odorants they bind to and may even assist in deorphanization of few of these ORs [43]. Evidence of more precise gene boundaries of ORs of closely related genomes will certainly improve OR prediction through homology-based annotation.

## Implementation

The core annotation pipeline, as described in the previous section, is invoked from the insectOR website. The webserver is written in Python with Jinja2 for templating and bootstrap framework which uses HTML, CSS and JavaScript/Ajax. Dalliance and its API is used for genome annotation visualization [37]. InsectOR also makes use of file conversion tools like faToTwobit [44], gff_to_bed.py (https://github.com/vipints/GFFtools-GX/blob/master/gff_to_bed.py) and bedToBigBed [45, 46] for visualization of the predictions. The basic insectOR package for standalone use can be downloaded from https://github.com/sdk15/insectOR.

## Precision and sensitivity calculations

We used 'gffcompare' [47] for comparison of insectOR gene predictions with predictions from other resources, detailed in further sections. Briefly, the method calculates sensitivity and precision at various levels of transcript annotation using standard formulae–

Sensitivity = TP / (TP + FN) and

Precision = TP / (TP + FP)

where, TP = True Positive, FP = False Positive, FN = False Negative

At the level of 'base', the number of the exon bases that are correctly predicted as part of the transcript (with respect to the reference transcript) is considered as TP, the number of reference exon bases not covered by the query transcripts are FN, whereas additional exon bases predicted by the query but absent in the reference are considered as FP. For the level of 'exon', unit of comparison is exon interval. At the level of 'Locus' at least one of the possible predicted transcripts should match one of the reference transcripts to be called as TP. In addition, we set parameter '-e' to 10 (more stringent) instead of default 100, which represents maximum distance that was allowed from free ends of terminal exons of reference transcript/s when calculating exon accuracy.

## Results and discussion

### Evaluation

We discuss the number of ORs we find in three insect genomes through InsectOR webserver in detail. Although a comparable webserver/method is not available for OR gene prediction

specifically, another general gene annotation pipeline (MAKER) was tested by providing comparable parameters for the first two genomes. MAKER was tuned for OR detection by specifying the maximum intron length of 2000 and by providing the same input query proteins for its Exonerate runs as provided for the corresponding insectOR runs. OR search in *Drosophila melanogaster* demonstrated the performance of our method on a well-annotated species. The second example demonstrated how the search for ORs in a blueberry bee (*H. laboriosa*) was made simpler and automatic, using the core pipeline that forms the basis of this webserver. Taking our own published final annotations of ORs from blueberry bee as a reference [20], the raw results from the current modified webserver and two other general annotation pipelines were compared. The performance of insectOR was compared with other methods for these two species as described below. InsectOR was more sensitive in predicting genes at base, exon and locus level as compared to MAKER and NCBI gene annotation. As a demonstration of the utility of this tool on a new, yet to be manually curated genome, we chose *Apis dorsata* and the predictions from insectOR and NCBI proteins are compared in further sections.

**Case study 1: *Drosophila melanogaster* ORs.** To test insectOR on a well-studied model organism, we chose fruit-fly *Drosophila melanogaster* genome (assembly Release 6 plus ISO1 MT) belonging to insect order Diptera.

The Ensembl reference gene annotations were taken as standard and only OR related information was retained. It possesses 61 OR gene loci encoding 65 OR mRNAs (including isoforms). For testing insectOR, the query protein dataset was built from well-curated 727 non-Drosophila OR protein sequences from NCBI non-redundant protein database belonging to the order Diptera (S1 File).

Exonerate [25] alignment of these proteins against the Drosophila genome was performed and it was provided as an input to insectOR. For de novo gene prediction within MAKER [48], two methods—AUGUSTUS 2.5.5 [49] and SNAP [50] were implemented. HMM gene model of 'aedes' was used for training AUGUSTUS and that of 'mosquito' was used for training SNAP de novo gene predictions as the gene models from the same non-'Drosophila' species were not available for the two methods. The predictions from insectOR and MAKER [48] were compared with those of the NCBI as reference using 'gffcompare' [47]. The results of the comparison are discussed in Table 1.

Out of the total 62 OR gene/fragments predicted by insectOR, 56 can be validated using 7tm_6 and they also show 99% base level precision, which means that almost all the OR gene loci are identified at correct locations. Fifty-five of these had length more than 300 amino acids. InsectOR showed better sensitivity at base, exon and locus levels. Some genes containing

**Table 1. *D. melanogaster* OR gene prediction assessment.**

| Reference mRNAs (Ensembl): 65 | | | | |
|---|---|---|---|---|
| OR prediction method | insectOR | | Maker | |
| No of predicted genes/gene-fragments | 62 | | 25 | |
| Proteins with one or more 7tm_6 predictions | 56 | | 24 | |
| Proteins with multiple 7tm_6 predictions | 0 | | 9 | |
| Missed exons | 8.6% | | 56.4% | |
| Missed loci | 1.60% | | 55.70% | |
| Matching loci | 35 | | 17 | |
| | Sensitivity | Precision | Sensitivity | Precision |
| Base | 87 | 99 | 43 | 84 |
| Exon | 74 | 76 | 37 | 78 |
| Locus | 57 | 61 | 28 | 68 |

ORs, predicted by insectOR, were not complete at the boundaries and hence it showed less precision at the exon and locus level, as compared to MAKER [48]. At the exon and locus level precision calculation, gffcompare method searches for exact matches (with only 10 bp allowed deviation at the boundaries) to be qualified for a true positive hit [47, 51]. However, this better precision at the exon and locus level for MAKER [48] was at the cost of sensitivity and it missed more than 50% of the OR gene loci completely. The output of gffcompare for *Drosophila melanogaster* is available in S2 File. This execution took around 3 hours to process Exonerate alignment file (9.1MB size containing 2099 alignments) on insectOR.

**Case study 2: *H. laboriosa* ORs.**   We evaluated performance of insectOR for a species from another insect order–Hymenoptera (includes bees, ants and wasps). As discussed before, the basis of this pipeline was developed during annotation of ORs from two solitary bees–*Habropoda laboriosa* (Blueberry bee) and *Dufourea novaeangliae*, of which we have compared *H. laboriosa* predictions here [20]. Compared to our previous analysis on *A. florea* ORs, which required manual intervention, we found significant extent of automation for the complete annotation of *H. laboriosa* using insectOR. When the final set of genes (coming from our complete semi-automated annotation) were compared with those from NCBI eukaryotic genome annotation pipeline (https://www.ncbi.nlm.nih.gov/genome/annotation_euk/Habropoda_laboriosa/100/) [52], significant improvement was observed in the coverage of the total number of OR genes and accuracy of gene models, as discussed. To summarize, after our complete semi-automated analysis, 42 completely new OR gene regions were found (27% of the total blueberry ORs found) as compared to the NCBI genome annotations. Eighty-two OR genes (54% of total blueberry ORs) already covered by NCBI gene annotations had serious problems with the gene and intron-exon boundaries that were corrected. An example of this is shown in Fig 1C where middle panel of 'User-uploaded-genes' shows prediction of ORs by NCBI annotation pipeline and the last panel shows predictions from insectOR. In this case, the NCBI gene annotation has predicted one fused gene model for four distinct OR gene loci, as it has failed to predict the last exon in each of these genes. Also, it has failed to identify the second gene region completely which is a pseudogene due to presence of an in-frame STOP codon TGA (as seen in the zoomed-in version–STOP codon is shown to translate into '*'). For more details on the number of novel and modified genes, please see the supplementary information in Karpe et. al., 2017.

Here we have compared raw OR gene predictions from insectOR (without further manual curation) with those from MAKER [48] and NCBI [52] (Table 2). The final manually curated gene predictions from the above-mentioned paper were taken as the reference. Briefly, published curated OR protein sequences from Hymenopteran species including *Apis mellifera*,

**Table 2.  *H. laboriosa* OR gene prediction assessment.**

| Reference mRNAs [20]: 151 | | | | | | |
|---|---|---|---|---|---|---|
| **OR prediction method** | **insectOR** | | **Maker** | | **NCBI** | |
| **No of predicted genes/gene-fragments** | 151 | | 133 | | 62 | |
| **Proteins with one or more 7tm_6 predictions** | 134 | | 92 | | 62 | |
| **Missed exons** | 13.9% | | 32.30% | | 49.30% | |
| **Missed loci** | 0.7% | | 15.30% | | 14.00% | |
| **Matching loci** | 57 | | 6 | | 14 | |
| | Sensitivity | Precision | Sensitivity | Precision | Sensitivity | Precision |
| **Base** | 87 | 95 | 73 | 85 | 54 | 80 |
| **Exon** | 65 | 68 | 31 | 40 | 33 | 55 |
| **Locus** | 38 | 39 | 4 | 5 | 9 | 23 |

*Apis florea*, *Bombus impatiens*, *Megachile rotundata*, *Lasioglossum albipes*, *Cerapachys biroi*, *Nasonia vitripennis*, *Microplitis mediator* and *Cephus cinctus* were collected. Fragmented (<100 amino acids) or extended (>600 amino acids) proteins were removed. Further sequences with only 7tm_6 domain identified with CD-search were retained (S3 File). These 1249 curated OR protein sequences (without self-OR sequences) were used as input for Exonerate within MAKER [48]. Similar to Drosophila, MAKER [48] annotations were carried out using de novo gene predictions from AUGUSTUS 2.5.5 [49] and SNAP [50], both trained on gene models from *A. mellifera*. In the raw output of our current insectOR webserver, 151 OR gene/gene-fragments were predicted. Out of these, 103 were complete (>300 amino acids in length) and 134 displayed presence of 7tm_6 domain. We could find only 133 OR proteins predicted by MAKER and only 62 by NCBI. Out of these 133 ORs predicted using MAKER, 65 were complete. But, 23 of the probable complete ones were more than 500 amino acids in length and were fused protein predictions indicating that providing similar maximum intron length cut-off for Exonerate was not enough for fine-tuning for OR gene prediction within MAKER. Similar fused proteins were observed for NCBI gene predictions. This is reflected in the number of proteins with multiple 7tm_6 domains from MAKER and NCBI. As shown in the Table 2, for all the measures of performance of the prediction, insectOR performed better than MAKER and NCBI annotations. This example is provided for sample execution at insectOR. The output of gffcompare for *Habropoda laboriosa* is available in S4 File. The sample execution took less than ten minutes to process Exonerate alignment file (45.9MB size containing 13180 alignments) on insectOR.

**Case study 3: *Apis dorsata* ORs.** As a third case study, we chose genome of giant honey bee, *Apis dorsata*. The genome of the species is published [53], but the manually curated list of ORs in not yet available. The number of published and curated OR sequences has increased since the curation of *Habropoda laboriosa* ORs, hence we made a more recent, comprehensive and representative dataset of 4354 OR queries (S5 File). These were used to perform Exonerate alignment and the resulting output was used as an input for insectOR as before. As manually curated OR gene regions were unavailable, we simply compared the insectOR prediction with those available at NCBI (Table 3). The expected number of ORs in *A. dorsata* is > = 165 given the discovery of ~165 olfactory glomeruli [54] as well as given the similar number of ORs in the other honey bees [16, 19].

Upon search in NCBI ('odorant receptor Apis dorsata') 128 protein sequences were found. Eighteen were marked as "uncharacterized protein". Although 11 of them had length smaller than 300, only 3 were marked as "partial". There were also 3 extended sequences which are difficult to split while performing manual curation. Of the total, 55 were marked as "isoform". In contrast, insectOR was able to predict more genes with 7tm_6. There were only 4 partial hits with 7tm_6 domain, which could be isoforms of the nearby complete genes, or genuinely partial copies. All 98 NCBI OR genes corresponding to 128 NCBI proteins were identified by 112

**Table 3. *A. dorsata* OR gene prediction assessment.**

| OR prediction method | InsectOR | NCBI proteins |
|---|---|---|
| No of predicted genes/gene-fragments | 270 | 128 (112 genes) |
| Proteins with 7tm_6 domain predictions | 174 | 128 |
| Complete hits with 7tm_6 (> = 300 aa) | 170 | 117 |
| Partial hits with 7tm_6 (<300 aa) | 4 | 11 |
| Extended hits with 7tm_6 (>500 aa) | 0 | 3 |
| Normal hits with 7tm_6 | 162 | 128 |
| Pseudogene hits with 7tm_6 | 12 | NA |

insectOR hits. As most of the insectOR hits are complete, this shows that approximately 14 gene regions predicted by NCBI actually span two or more complete 7tm_6 coding gene regions and need to be split. At the same time, it can be confirmed that insectOR predicted 62 (174–112) novel gene regions coding for 7tm_6 containing proteins / insect olfactory receptors.

Here, it will be easier to acquire all and mostly complete genes if users start from insectOR annotations than NCBI annotations for further manual curation. Hits with 7tm_6 domains provide a rough estimate of total OR genes to expect from the genome. Recommended steps after insectOR are– 1) Manually curate the ORs that are already marked as complete and contain 7tm_6, checking mainly the 5'and 3'ends. If START or STOP codons are missing, nearby partial hits without 7tm_6 domain could be extensions of such 'complete' hits. 2) Pseudogenes can be checked to see whether there are alternate partial hits available nearby that could provide a complete and non-pseudogenous structural annotation. 3) Next target the partial but 7tm_6 containing hits and check whether nearby partial and without 7tm_6 containing hits could complete them. 4) The remaining sequences could be checked for similarity with already manually curated ORs to check whether there is a high identity amongst the two sets. The sequences with low identity are truly false positives and can be discarded.

Furthermore, we applied InsectOR on four other insect genomes and these results are organized in S1-S4 Tables in S6 File. The numbers of 7tm_6 containing hits as predicted by insectOR are similar to the total expected number of ORs for these genomes.

## Conclusion

InsectOR is a first-of-a-kind webserver for the prediction of ORs from newly sequenced genomes of insect species. Insect OR genes are diverse across various taxonomical categories and hence these are hard to detect for general genome annotation pipelines, which also tend to wrongly predict fused tandem OR gene models. InsectOR outperforms such general genome annotation methods in providing accurate gene boundaries, reducing the efforts spent on manual curation of this huge family of proteins. Overall, InsectOR performed well across two different insect orders and provided best sensitivity and good precision amongst the methods tested here for OR gene prediction.

InsectOR performance is dependent on the initial query set. Wherever possible, it is recommended to employ curated query/reference OR sequences from organisms which are known to be evolutionarily closely related to the species of interest to ensure good sensitivity. On the other hand, inclusion of non-OR sequences in the reference set might lead to large number of false positives. The users can detect false positives mainly by looking at the insectOR output table and identifying the near complete normal (>300 amino acids) genes coding for proteins without '7tm_6' Pfam domain or any transmembrane helices or known motifs. If many of such hits arose from few of the same 'best' query sequences as notified in the table, that specific query should be ideally evaluated to check whether it is indeed an OR. Though InsectOR annotations are not yet complete for few genes near the gene-boundaries, it displays the relevant information showing whether each gene is incomplete or pseudogenous. Unfortunately, true partial and pseudogenous OR gene fragments may not show presence of the above characteristics and hence cannot be filtered out at this time. These could be fragments or alternative isoforms of nearby near-complete genes. Hence, the absence of 7tm_6 cannot be confidently interpreted as false positive. Further measures (limited manual editing or expression analysis) can be taken by the user to ensure completeness of these models. The number of 7tm_6 Pfam domain containing genes reported by insectOR is a good approximation of the total number of true positive ORs being predicted by the tool.

With current ongoing projects of sequencing 1000s of insect genomes and transcriptomes, the webserver and the standalone package has potential to serve many entomologists all over the world. We believe, it will reduce the overall time taken for final manual curation of OR genes, to about one-fourth, of the usual from our previous experience. It is a first step towards annotation methods tuned for huge protein families like ORs and in the future it could be adapted to other similar diverse protein families.

## Supporting information

**S1 File. The query/reference OR proteins used for OR prediction from *Drosophila melanogaster* genome.**
(PDF)

**S2 File. The output of gffcompare for *Drosophila melanogaster*.** Detailed result of comparison of gene annotations by MAKER and insectOR to the NCBI annotations as reference for the *Drosophila melanogaster* genome.
(PDF)

**S3 File. The query/reference OR proteins used for OR prediction from *Habropoda laboriosa* genome.**
(PDF)

**S4 File. The output of gffcompare for *Habropoda laboriosa*.** Detailed result of comparison of gene annotations by MAKER, NCBI and insectOR to the curated annotations as reference for the *Habropoda laboriosa* genome.
(PDF)

**S5 File. The query/reference OR proteins used for OR prediction from *Apis dorsata* genome.**
(PDF)

**S6 File. Application of InsectOR webserver on other insect species.**
(PDF)

## Acknowledgments

We would like to thank, Mr. Murugavel Pavalam for extensive help with improving the core algorithm of insectOR and also for helping to implement it on the web platform. We would like to thank National Centre for Biological Sciences (NCBS) for infrastructural facilities.

## Author Contributions

**Conceptualization:** Sowdhamini Ramanathan.

**Data curation:** Vikas Tiwari.

**Formal analysis:** Vikas Tiwari.

**Investigation:** Vikas Tiwari.

**Methodology:** Snehal Dilip Karpe.

**Project administration:** Snehal Dilip Karpe, Sowdhamini Ramanathan.

**Software:** Snehal Dilip Karpe.

**Supervision:** Sowdhamini Ramanathan.

**Validation:** Vikas Tiwari.

**Visualization:** Snehal Dilip Karpe.

**Writing – original draft:** Snehal Dilip Karpe, Vikas Tiwari.

**Writing – review & editing:** Sowdhamini Ramanathan.

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
