## [Decision Letter · Decision Letter 0]

9 Sep 2020

PONE-D-20-12954

InsectOR - webserver for sensitive identification of insect olfactory receptor genes from non-model genomes

PLOS ONE

Dear Dr. Sowdhamini,

Thank you for submitting your manuscript to PLOS ONE. After careful consideration, we feel that it has merit but does not fully meet PLOS ONE’s publication criteria as it currently stands. Therefore, we invite you to submit a revised version of the manuscript that addresses the points raised during the review process.

Please read through the comments of the two referees but also include the following two additions:

- Address the issues of false positives, as explained by reviewer 1. Take attention on your response as this will ensure that your tool is used and the paper is read and cited.

- Include a case study of a species that has not been studied before to show how can this software be used

- Explain whether the software can be downloaded and used off-line by bioinformaticians and whether the code is open source. If not, then you may wish to consider to do that as it is unlikely the community will use something they cannot validate themselves.

We look forward to receiving your revised manuscript.

Kind regards,

Alexie Papanicolaou, PhD

Academic Editor

PLOS ONE

Journal Requirements:

Reviewers' comments:

Reviewer's Responses to Questions

**Comments to the Author**

1. Is the manuscript technically sound, and do the data support the conclusions?

Reviewer #1: Yes

Reviewer #2: Yes

2. Has the statistical analysis been performed appropriately and rigorously? 

Reviewer #1: Yes

Reviewer #2: Yes

3. Have the authors made all data underlying the findings in their manuscript fully available?

Reviewer #1: Yes

Reviewer #2: Yes

4. Is the manuscript presented in an intelligible fashion and written in standard English?

Reviewer #1: Yes

Reviewer #2: Yes

5. Review Comments to the Author

Reviewer #1: The authors have created a web tool for semi-automatic identification and annotation of insect Odorant Receptor genes from uncharacterized genomic sequences. This tool would be useful for biologists that lack the bioinfomartics background to perform these queries manually (this reviewer’s perspective). The tool would allow for relatively quick surveys of important olfactory genes in non-model species. These data could complement other types of investigations: behavioral, ecological, etc. A bioinformaticist is needed to validate all methodology (for instance the parameters used for the MAKER comparison, line 226), but the techniques appear well intentioned and designed. The rationale for each case study species makes sense, though I had hoped a completely unstudied species was used for case study #2 to demonstrate the potential power of this web tool.

Main Comments:

Perhaps the abstract should end with a statement reminding readers of this tool’s value to researchers.

The paper does not fully address this web tool’s false positive rate of discovery. Several of the trials run in tables S1-S4 show too many gene hits as compared to what has currently been uncovered. It could be that these hits are undiscovered ORs, but it is likely some or most of these extra hits are false positives. A section needs to be added to the methods that describes how precision and sensitivity were measured, giving more details. The conclusion needs to discuss how to determine false positives in the output files.

Line 234 – Be more specific as to how insectOR performed “better.”

Line 242 – it would be helpful to include a file listing the 727 non-drosophila ORs used in this step

Table 1 – I did not notice a thorough explanation of how “sensitivity” and “precision” were calculated in the case study. The accompanying text is somewhat vague.

Line 291 – rather than referencing the 2017 paper, it would be helpful to include a file listing the 1249 ORs used in this step, or at least a brief explanation of how these were chosen

Line 307 – change “five other genomes” to “four other genomes”

Small corrections by line number:

16 – delete “Hence”

19 – change “sever” to “server” in abstract

39 – remove capital in “Olfaction”

48 – “is” to “are”

56 – delete “only”

285 – change “missed” to “failed”

321-reword?

Reviewer #2: The manuscript by Karpe et al. develops a webserver – InsectOR serving as the prediction of gene models of insect ORs. This is a significant work for the preliminary identification of OR genes. This manuscript was well written and easy to be followed. However, I am not an expert in the field of bioinformatics. For the manuscript, I mainly concerned two issues:

1. Insect ORs are a diverse family with some genes exhibiting extremely low identities at the amino acid levels. As I know, some of ORs in one species have no orthologs to ORs from other species. This is difficult for the researchers to identify the exact number or full-length sequences of OR genes. Whether the authors have considered the parameter set of this webserver on low identities or no orthologs of ORs?

2. If one gene was located on two scaffolds or contigs, this webserver enables them merge into one gene?

3. Using ‘InsectOR’, this number of genes is relatively precise. I would like to know whether it will give more full-length sequences?

6. PLOS authors have the option to publish the peer review history of their article (what does this mean?). If published, this will include your full peer review and any attached files.

Reviewer #1: No

Reviewer #2: No

---

## [Author Response · Author response to Decision Letter 0]

5 Nov 2020

Editor’s comments -

1) Address the issues of false positives, as explained by reviewer 1. Take attention on your response as this will ensure that your tool is used and the paper is read and cited.

We have addressed the relevant reviewer question below and also added information to methods and conclusion section as requested by the reviewers.

2) Include a case study of a species that has not been studied before to show how can this software be used

We have now added case study 3 with insectOR gene predictions for Apis dorsata genome and provided instructions about further manual curation as well

3) Explain whether the software can be downloaded and used off-line by bioinformaticians and whether the code is open source. If not, then you may wish to consider to do that as it is unlikely the community will use something they cannot validate themselves.

We have now uploaded the tool to GitHub and it is open source. It is available at -

https://github.com/sdk15/insectOR

Comments to the Author

Review Comments to the Author

1) Reviewer #1: The authors have created a web tool for semi-automatic identification and annotation of insect Odorant Receptor genes from uncharacterized genomic sequences. This tool would be useful for biologists that lack the bioinformatics background to perform these queries manually (this reviewer’s perspective). The tool would allow for relatively quick surveys of important olfactory genes in non-model species. These data could complement other types of investigations: behavioral, ecological, etc. A bioinformatician is needed to validate all methodology (for instance the parameters used for the MAKER comparison, line 226), but the techniques appear well intentioned and designed. The rationale for each case study species makes sense, though I had hoped a completely unstudied species was used for case study #2 to demonstrate the potential power of this web tool.

We have now added case study 3 with insectOR gene predictions for Apis dorsata genome.

2) Perhaps the abstract should end with a statement reminding readers of this tool’s value to researchers.

Thank you for this suggestion. We have now added a sentence accordingly.

3) The paper does not fully address this web tool’s false positive rate of discovery. Several of the trials run in tables S1-S4 show too many gene hits as compared to what has currently been uncovered. It could be that these hits are undiscovered ORs, but it is likely some or most of these extra hits are false positives. A section needs to be added to the methods that describes how precision and sensitivity were measured, giving more details. The conclusion needs to discuss how to determine false positives in the output files.

Thank you for these suggestions. We have now added relevant information in the methods and the conclusion sections. We have also added a new publication reference [47] for ‘gffcompare’ utility which only existed as a webpage at the time of previous submission.

More explanation to reviewer points is given here -

The total number of genes/gene regions identified by insectOR cannot be expected to be close to the total number of insect OR genes in the given genome. This is simply because there could be many partial genes, few of which are also isoforms of the nearby complete genes. InsectOR prefers complete and more compact gene models over other overlapping gene predictions. Hence other isoforms are left as partial. This cannot be resolved without further information from antennal transcriptome data. In case of draft and segmented genome assemblies, split OR genes cannot be merged by insectOR.

Better measure of the total number of OR genes is the number of 7tm_6 containing OR protein coding genes and it is a good indicator of True Positives. On the contrary, the absence of 7tm_6 cannot be confidently interpreted as a False Positive for the following reasons –

a) 7tm_6 HMM was initially built using only Drosophila melanogaster olfactory receptors. As more olfactory receptors are sequenced, it is becoming clearer that certain OR subfamilies also seem to be exclusively present in certain insect taxonomic orders (Hansson & Stensmyr 2011, Karpe SD Ph. D. Thesis 2018). Hence ORs that are distantly related to Drosophila ORs might fail to show presence of this domain using HMMSEARCH.

b) The presence of the Pfam domain identified here also depends on the length of the protein provided, with partial ones smaller than a certain length failing the identification altogether. 

c) In case of pseudogenes and their translated pseudo-proteins (mainly used for phylogenetic reconstructions), key amino acids defining 7tm_6 domains might be missing, again leading to no 7tm_6 identification.

Hence, we do not use this information to either filter out any hits or to calculate false discovery rate. We choose to rather inform the users to use it for their benefit while stitching closely spaced partial/pseudo genes, if any. Users can also perform additional analyses looking at the sequence identity with well-known olfactory receptors to decide the fate of gene regions without any identified domain.

Next, the prediction by insectOR depends largely on the query sequences used for performing Exonerate. Users are advised to choose these sequences with care. More instructions are given in the ‘How to choose OR protein queries for Exonerate alignment?’ section of the ‘About’ tab of insectOR webserver. Although utmost care was taken while performing the analyses on four additional genomes, we do still see high number of genes predicted. Many of these can be explained by the reasons provided above. Few of these hits can be refined further using the additional information provided by insectOR. This process is difficult to automate without losing sensitivity and defeats the purpose of insectOR, which is to provide preliminary identification of olfactory receptors.

4) Line 234 – Be more specific as to how insectOR performed “better.”

We have rewritten this sentence now.

5) Line 242 – it would be helpful to include a file listing the 727 non-drosophila ORs used in this step

We have included this file in the supplementary data.

6) Table 1 – I did not notice a thorough explanation of how “sensitivity” and “precision” were calculated in the case study. The accompanying text is somewhat vague.

We have now added a paragraph in methods section for better understanding.

7) Line 291 – rather than referencing the 2017 paper, it would be helpful to include a file listing the 1249 ORs used in this step, or at least a brief explanation of how these were chosen

We have added explanation regarding these and also included the file in the supplementary data.

8) Line 307 – change “five other genomes” to “four other genomes”

We have changed the number of the genomes.

9) Small corrections by line number:

16 – delete “Hence”

19 – change “sever” to “server” in abstract

39 – remove capital in “Olfaction”

48 – “is” to “are”

56 – delete “only”

285 – change “missed” to “failed”

321-reword?

We have made the corrections suggested here. For the change on line 48, we have changed ‘is’ to ‘belong to’ as it seemed more appropriate.

Reviewer #2: The manuscript by Karpe et al. develops a webserver – InsectOR serving as the prediction of gene models of insect ORs. This is a significant work for the preliminary identification of OR genes. This manuscript was well written and easy to be followed. However, I am not an expert in the field of bioinformatics. For the manuscript, I mainly concerned two issues:

10) Insect ORs are a diverse family with some genes exhibiting extremely low identities at the amino acid levels. As I know, some of ORs in one species have no orthologs to ORs from other species. This is difficult for the researchers to identify the exact number or full-length sequences of OR genes. Whether the authors have considered the parameter set of this webserver on low identities or no orthologs of ORs?

It is true that many olfactory receptors do not have orthologs in other species, however their homologs do exist at lower sequence identities. It has been observed that the ORs from various insect orders tend to cluster separately (Hansson & Stensmyr 2011). However, many more insect genomes and antennal transcriptomes have been sequenced since then and now representative ORs from almost all insect taxonomic orders and many taxonomic families are available. In our experience with insectOR, including as diverse sequences of ORs (mainly from the same insect order) helps to improve the number as well as completeness of the genes predicted, clearly due to higher sequence identities. More instruction about selecting query sequences for insectOR run are provided in the ‘How to choose OR protein queries for Exonerate alignment?’ section of the ‘About’ tab of insectOR. 

However, if users feel that the numbers are still unsatisfactory, they can choose to use PAM250 distance matrix while performing Exonerate alignment. This matrix is suited to identify more distantly related / homologous sequences. More details are provided in the ‘How to generate Exonerate alignment file?’ section of the ‘About’ tab of insectOR. This increases the sensitivity of the search, but it also increases false positives. Hence users need to be careful while analysing results from this search. An alternative to this would be to take output proteins of the first insectOR run, filter them to identify the near-complete 7tm_6 domain containing sequences and use them for the second round of Exonerate-insectOR. Users have reported that this helped them to identify more OR genes.

At the level of insectOR algorithm, we do not filter out any potential gene region identified by Exonerate+GeneWise, unless it is overlapped by another more complete gene prediction. We report all of them along with their validations including the presence of 7tm_6 domain, presence of known OR motifs and presence of transmembrane helices. Users can further filter out certain regions which do not pass their own cutoffs for these validations.

11) If one gene was located on two scaffolds or contigs, this webserver enables them merge into one gene?

InsectOR provides information of the ‘best’ query sequence that helped to define the genomic location of every predicted gene. This information is available in the tabular output. Although insectOR cannot directly merge a split gene on two scaffolds, users can make use of the information about the shared best query and location of the two partial genomic regions on respective two scaffolds. If they are located near the ends of the two scaffolds in question and two distinct regions of the best query align to them, it is likely that it is a split gene. It should be also noted that there can be duplication of genes in the genome and the same ‘best’ query might identify two such complete or partial regions. Hence this needs to be done cautiously. Currently, it is impossible to confidently merge partial genes on two different scaffolds without use of additional information, say antennal transcriptome assembly data or an improved genome assembly.

12) Using ‘InsectOR’, this number of genes is relatively precise. I would like to know whether it will give more full-length sequences?

We use GeneWise for the second level predictions, which helps to improve the number of complete genes significantly than just processing the results from Exonerate alone (P1 predictions). As we did not find de novo or homology-based predictions from tools like MAKER to be comparable in the performance, we do not think that any additional tool might help in this aspect. 

Secondly, it should be noted that the insectOR gives preference to genes that are most compact and complete. Hence, some of the partial genes/predicted regions might reflect exons belonging to alternative isoforms of the nearby complete genes predicted by insectOR. In the absence of the transcriptome data, it is impossible to declare the isoforms at this stage, and hence they must be declared as a separate partial gene at this time.

Thirdly, if the genome assembly is fragmented, it might again lead to multiple partial hits. As noted in the previous answer, a bit of additional work on user’s part might help to merge split genes on two scaffolds.

Lastly, some users have reported that complete 7tm_6 containing output sequences from insectOR can be used for a second round of prediction with Exonerate-insectOR. This might help to improve the number and completeness of genes.

---

## [Decision Letter · Decision Letter 1]

29 Dec 2020

InsectOR - webserver for sensitive identification of insect olfactory receptor genes from non-model genomes

PONE-D-20-12954R1

Dear Dr. Sowdhamini,

We’re pleased to inform you that your manuscript has been judged scientifically suitable for publication and will be formally accepted for publication once it meets all outstanding technical requirements.

Kind regards,

Alexie Papanicolaou, PhD

Academic Editor

PLOS ONE

Additional Editor Comments (optional):

Reviewers' comments:

Reviewer's Responses to Questions

**Comments to the Author**

1. If the authors have adequately addressed your comments raised in a previous round of review and you feel that this manuscript is now acceptable for publication, you may indicate that here to bypass the “Comments to the Author” section, enter your conflict of interest statement in the “Confidential to Editor” section, and submit your "Accept" recommendation.

Reviewer #1: All comments have been addressed

Reviewer #2: All comments have been addressed

2. Is the manuscript technically sound, and do the data support the conclusions?

Reviewer #1: Yes

Reviewer #2: Yes

3. Has the statistical analysis been performed appropriately and rigorously? 

Reviewer #1: Yes

Reviewer #2: Yes

4. Have the authors made all data underlying the findings in their manuscript fully available?

Reviewer #1: Yes

Reviewer #2: Yes

5. Is the manuscript presented in an intelligible fashion and written in standard English?

Reviewer #1: Yes

Reviewer #2: Yes

6. Review Comments to the Author

Reviewer #1: All changes made in response to first review have been well executed. Thank you for your thorough revision. Particularly helpful are the third case study and the added section explaining 'Precision and Sensitivity.' This works provides a very useful tool to chemical ecologists.

Reviewer #2: The authors have responded to my concerns, and I have no additional comments for this manuscript. Thus, I recommend a acception in PLoS One.

---

## [Editor Report · Acceptance letter]

6 Jan 2021

PONE-D-20-12954R1 

InsectOR – webserver for sensitive identification of insect olfactory receptor genes from non-model genomes 

Dear Dr. Ramanathan:

I'm pleased to inform you that your manuscript has been deemed suitable for publication in PLOS ONE. Congratulations! Your manuscript is now with our production department. 

Kind regards, 

on behalf of

Dr. Alexie Papanicolaou 

Academic Editor

PLOS ONE